# Finely-Tuned Calcium Oscillations in Osteoclast Differentiation and Bone Resorption

**DOI:** 10.3390/ijms22010180

**Published:** 2020-12-26

**Authors:** Hiroyuki Okada, Koji Okabe, Sakae Tanaka

**Affiliations:** 1Department of Orthopaedic Surgery, The University of Tokyo, Tokyo 113-8655, Japan; hokada-tky@umin.ac.jp; 2Center of Disease Biology and Integrative Medicine, Graduate School of Medicine, The University of Tokyo, Tokyo 113-8655, Japan; 3Department of Physiological Science and Molecular Biology, Fukuoka Dental College, Fukuoka 814-0193, Japan; okapi@college.fdcnet.ac.jp

**Keywords:** calcium oscillation, osteoclast, receptor activator of nuclear factor kappa B ligand (RANKL), transient receptor potential (TRP) channel, costimulatory signal, immunoreceptor tyrosine-based activation motif (ITAM)

## Abstract

Calcium (Ca^2+^) plays an important role in regulating the differentiation and function of osteoclasts. Calcium oscillations (Ca oscillations) are well-known phenomena in receptor activator of nuclear factor kappa B ligand (RANKL)-induced osteoclastogenesis and bone resorption via calcineurin. Many modifiers are involved in the fine-tuning of Ca oscillations in osteoclasts. In addition to macrophage colony-stimulating factors (M-CSF; CSF-1) and RANKL, costimulatory signaling by immunoreceptor tyrosine-based activation motif-harboring adaptors is important for Ca oscillation generation and osteoclast differentiation. DNAX-activating protein of 12 kD is always necessary for osteoclastogenesis. In contrast, Fc receptor gamma (FcRγ) works as a key controller of osteoclastogenesis especially in inflammatory situation. FcRγ has a cofactor in fine-tuning of Ca oscillations. Some calcium channels and transporters are also necessary for Ca oscillations. Transient receptor potential (TRP) channels are well-known environmental sensors, and TRP vanilloid channels play an important role in osteoclastogenesis. Lysosomes, mitochondria, and endoplasmic reticulum (ER) are typical organelles for intracellular Ca^2+^ storage. Ryanodine receptor, inositol trisphosphate receptor, and sarco/endoplasmic reticulum Ca^2+^ ATPase on the ER modulate Ca oscillations. Research on Ca oscillations in osteoclasts has still many problems. Surprisingly, there is no objective definition of Ca oscillations. Causality between Ca oscillations and osteoclast differentiation and/or function remains to be examined.

## 1. Introduction

Calcium (Ca^2+^) is a simple molecule, but has various cellular functions [1]. It acts as a common second messenger in many biological process [2,3]. For example, small changes in Ca^2+^ can induce dynamic cellular functions, including synapse transduction in neural cells [4], muscle contraction [5], and fertilization in oocytes [6].

Hematopoietic stem cell-derived osteoclasts are electrically stable cells, and the Ca^2+^ concentrations in bone marrow macrophages (BMMs) and osteoclasts are maintained at almost constant levels. However, subtle changes in the Ca^2+^ levels with and without extracellular stimuli, so-called calcium oscillations (Ca oscillations), play important roles in the cellular differentiation, function, and death of osteoclasts.

The effect of Ca oscillations on the differentiation process of osteoclasts remains under debate. Receptor activator of nuclear factor kappa B ligand (RANKL) is an essential inducer of osteoclast differentiation [7]. Nuclear factor of activated T-cells 1 (NFATc1) is a master regulator gene for osteoclastogenesis [8]. In addition to these essential factors for osteoclastogenesis, appropriate Ca oscillations are necessary depending on the situation for RANKL-induced NFATc1 auto-amplification.

In osteoclast differentiation, calcineurin is an important activator of NFATc1 converting Ca oscillations signals under RANKL transduction pathway [9]. Recent study clarified PICK1 is a positive regulator of calcineurin B [10]. Another group showed that mTORC1 impedes NFATc1 activation by calcineurin [11]. Cyclosporine A, which is an inhibitor of calcineurin, inhibited bone resorption by osteoclast in vitro [12,13], however, another group reported cyclosporine A did not affect bone resorption significantly [14]. The role of calcineurin in bone resorption might be controversial.

In this review, we will explain comprehensively important parts for inducing intracellular Ca oscillations (Figure 1). Initially, we will focus on costimulatory signals of osteoclastogenesis in the upstream of Ca oscillations. Next, we will discuss Ca^2+^ channels, especially transient receptor potential (TRP) channels. Intracellular storage also plays an important role in fine-tuning of Ca oscillations. Furthermore, environmental factors around osteoclasts affect intracellular Ca^2+^ concentration. Finally, we will present research perspectives on Ca oscillations in osteoclasts.

## 2. Costimulatory Signals during Osteoclast Development

Appropriate Ca oscillations are necessary for NFATc1 auto-amplification in osteoclastogenesis [8]. Koga et al. reported that immunoreceptor tyrosine-based activation motif (ITAM)-harboring adaptors transduced costimulatory signals during osteoclastogenesis. ITAM receptor-adaptor complexes acted as key modulators of Ca oscillations in BMMs [15]. Phosphorylated ITAM adaptor proteins recruit Syk tyrosine kinase and induce Ca oscillations through activation of PLCγ [16]. Syk, c-Src, and αvβ3 integrin cooperate under costimulatory signaling [17]. Protein tyrosine kinase inhibitors disrupt actin organization and osteoclast activity [18].

Monocytes and BMMs have two different types of ITAM adapter proteins, DNAX activating protein of 12 kD (DAP12) and Fc receptor gamma (FcRγ). These two types of ITAM proteins have different roles in osteoclastogenesis.

Measurement methods of Ca oscillations in Section 2 were summarized on Table 1 in order of description.

### 2.1. DAP12

DAP12 was reported as a disease gene for Nasu–Hakola disease, and DAP12 KO mice show increased bone mass (osteopetrosis) [22]. Under DAP12 depletion, Ca oscillations are deactivated in osteoclast precursor cells [15]. The homolog DAP10 regulates osteoclastogenesis by cooperating with myeloid DAP12-associating lectin-1 (MDL-1) [23].

TREM-2 is a receptor associated with DAP12 on myeloid cells [24]. TREM-2 regulates osteoclast differentiation and function. In clinical settings, TREM-2-deficient patients exhibit increased immature osteoclasts and impaired bone resorption [25]. Interestingly, TREM-2 gene expression is not regulated by NFATc1 [26], and the anti-inflammatory cytokine interleukin-10 inhibits TREM-2 expression and costimulatory signals in osteoclasts [19].

Siglec proteins are plasmalemmal receptors that recognize sialylated glycans. Siglec-15, a receptor associated with DAP12, regulates osteoclast differentiation [27], and Siglec-15-deficient mice show osteopetrosis [28]. As DAP12-associated receptors, TREM-2 and Siglec-15 have important roles in osteoclastogenesis. However, the different roles for Ca oscillations among the receptors and DAP12 remain unclear.

### 2.2. FcRγ

In contrast to DAP12 and its coupling receptors, the role of FcRγ in osteoclastogenesis is ambiguous and environmentally dependent. Grevers et al. reported that Fcγ receptor activation by immune complexes inhibited osteoclastogenesis [29]. Meanwhile, Seeling et al. reported that IgG autoantibody binding to Fcγ receptors promoted osteoclast differentiation and bone resorption [30]. Moreover, Negishi-Koga et al. reported that osteoclastogenesis is regulated in accordance with the environmental inflammatory state. In the physiological state, FcRγ couples with PIR-A or OSCAR and promotes osteoclastogenesis. While, in the pathological inflammatory situation with an abundance with immune complex, FcRγ binds to several types of Fcγ receptors. FcγR I/III/IV strengthen FcRγ signaling, in contrast, FcγRIIB weaken FcRγ signaling. The coupling patterns of Fcγ receptors with FcRγ modify the strength of FcRγ signaling and affect osteoclastogenesis [20].

Our group showed that a rheumatic drug, cytotoxic T-lymphocyte antigen 4 (CTLA4)-Ig, inhibits osteoclastogenesis by interfering with Ca^2+^ signaling via FcRγ, representing the first report of a cofactor that affects costimulatory signals during osteoclast differentiation [21]. Osteoclastogenesis may be finely-tuned via FcRγ with immunoglobulins and related cofactors according to immunological situations.

DAP12 delivers costimulatory signals in a direct and straightforward manner during osteoclast differentiation. In contrast, the costimulatory signals through FcRγ are complicated and environment-dependent. It remains to be solved why the downstream mechanisms including Ca oscillations differ according to ITAM types.

## 3. Calcium Channels and Transporters in Osteoclast

Ca oscillations under costimulatory signal are necessary for osteoclastogenesis [31]. However, receptors of costimulatory signals themselves do not affect directly intracellular Ca concentration like Ca channels or transporters. Ca oscillations are evoked by membranous and intracellular organs in a coordinated manner. In this section, plasmalemmal components inducing Ca oscillations are discussed.

Measurement methods of Ca oscillations in Section 3 were summarized on Table 2.

### 3.1. TRP Family

Some ionic channels on osteoclasts are activated by Ca^2+^, voltage, and even cellular stretching [51]. In addition, outside fluid flow induces different Ca^2+^ alterations according to the osteoclast differentiation stage [52].

TRP channels are well-known to work as environmental sensors for factors such as environmental pressure, acid, taste, and temperature [53]. The TRP family members highly contribute to osteoclast differentiation and function.

TRP vanilloid 1 (TRPV1) was identified as a capsaicin receptor [54] and is sensitive to heat [55]. In TRPV1 knockout (KO) mice, osteoclast differentiation is attenuated by decreasing Ca oscillations. Osteoblast differentiation is also disrupted and fracture healing is delayed [56]. Pharmacological blockade of TRPV1 channels inhibits osteoclast and osteoblast differentiation, and alleviates bone loss induced by ovariectomy [57] and tail suspension [58].

TRPV2 is a 50% homolog of TRPV1 that mediates high-threshold noxious heat sensation [59]. RANKL induces TRPV2 expression, activates Ca oscillations, and induces osteoclastogenesis through Ca^2+^-NFAT pathway [32]. In multiple myeloma, TRPV2 enhanced Ca^2+^-calcineurin-NFAT signaling [33].

TRPV4, an approximate 40% homolog of TRPV1, transduces warm stimuli [60]. TRPV4 cooperates with STIM1 and mediates fluid flow-induced Ca oscillations in osteoclast differentiation [34]. TRPV4 induces Ca^2+^ influx, activates calmodulin signaling, and regulates late differentiation of osteoclasts [35,36]. TRPV4 depletion suppresses osteoclastogenesis through the Ca^2+^-calcineurin-NFAT pathway [61].

TRPV5 and TRPV6 are homomeric and heteromeric epithelial channels that exhibit the highest Ca^2+^ selectivity among the TRP channels [62]. TRPV5 mediates RANKL-induced intracellular Ca^2+^ increases and reduces bone resorption due to a negative feedback mechanism to reduce the bone resorptive activity of mature osteoclasts [37]. Estrogen increases TRPV5 expression and inhibits osteoclast differentiation [38]. Although TRPV6 is abundant in bone cells, it is not crucial for mineralization [63]. Meanwhile, TRPV6 depletion promotes osteoclastic differentiation and function, and results in osteopenia [64].

Within the TRP family, TRPV channels are highly involved in osteoclast differentiation and function. Focusing on TRP family members other than TRPV, TRP canonical 1 (TRPC1) [65], TRPC3, and TRPC6 [39] regulate Ca^2+^ storage in osteoclasts.

### 3.2. Voltage-Gated Ca^2+^ Channels

The plasmalemmal voltage altered the activities of some ionic channels, including Ca^2+^ channels, in electrophysiological experiments in osteoclasts [40,51]. For example, the T-type Ca^2+^ channel Cav3.2, a target of the anticonvulsant drug diphenylhydantoin, positively regulates Ca signaling, NFATc1 activation, and osteoclastogenesis [41].

Voltage-gated Ca^2+^ channels also control osteoclast podosome formation and bone resorption [42]. Electrical Ca^2+^ entry and store refilling also define osteoclast survival [66].

Some Ca^2+^ channel modulators alter osteoclast function. Ca^2+^ channel agonists open Ca^2+^ channels on osteoclasts and decrease bone resorption [67]. Intracellular elevation of cytosolic Ca^2+^ induces osteoclast migration [68].

Ca^2+^ channels are modulated by certain plasmalemmal proteins. For example, regulator of G protein signaling 12 (RGS12) is involved in late differentiation of osteoclasts by Ca oscillations via N-type Ca^2+^ channels [43]. RGS12 promotes osteoclastogenesis and results in pathological bone loss [44]. RGS12 also controls osteoblast differentiation via Ca oscillations and the Gαi-ERK pathway [69]. RGS10 is necessary for Ca oscillations, NFATc1 signaling, and osteoclastogenesis [45].

### 3.3. K^+^ Channels

High extracellular Ca^2+^ and H^+^ induce voltage-gated outward efflux of potassium (K^+^) [70]. The Ca^2+^-dependent K^+^ current activates osteoclast spreading kinetics [71]. In a recent paper, the Ca^2+^-activated K^+^ channel KCa3.1 is shown to regulate Ca^2+^-dependent NFATc1 expression in the inflammatory situation [46].

Meanwhile, K^+^ channels can restrict osteoclast differentiation. K^+^ channel subfamily K member 1 (KCNK1) inhibits osteoclastogenesis by blocking Ca oscillations [47]. High-K^+^ solution depolarizes the osteoclast membrane potential through K^+^ channels. As a result, the driving force for Ca^2+^ influx into the cells is diminished, and the intracellular Ca^2+^ concentration decreases [48].

### 3.4. Ca^2+^-ATPase and Na^+^-Ca^2+^ Exchanger

Ca^2+^-ATPase is a calcium transporter associated with ATP hydrolysis. Ca^2+^-ATPase regulates bone mass in vivo through osteoclast differentiation and survival. Ca^2+^-ATPase inhibitors increase intracellular Ca^2+^ and induce osteoclast formation in a coculture system [72]. Plasmalemmal Ca^2+^-ATPase maintains bone mass by reducing Ca oscillations and limiting osteoclast differentiation and survival [49]. Meanwhile, Na^+^-Ca^2+^ exchanger (NCX) is an active transporter that excretes Ca^2+^ extracellularly in exchange for Na^+^ uptake. NCX1 and NCX3 are expressed in mature osteoclasts and significantly increase the intracellular Ca^2+^ concentration by removing extracellular Na^+^ [50].

## 4. Intracellular Calcium Storage in Osteoclast

In addition to Ca^2+^ exchange with the extracellular domain, intracellular organelles that store Ca^2+^ are involved in cytoplasmic alterations to the intracellular Ca^2+^ concentration [66]. Ca^2+^ store refilling is closely related to osteoclast function. On electrochemical microscopy, bone-resorbing osteoclasts show intracellular functional Ca^2+^ compartments [73]. In this section, we will focus on the internal storehouse of Ca.

Measurement methods of Ca oscillations in Section 4 are summarized in Table 3.

### 4.1. Endoplasmic Reticulum

The largest intracellular Ca^2+^ storage organelle is the endoplasmic reticulum (ER). Several calcium receptors on the ER membrane modulate the cytosolic Ca^2+^ concentration. Ryanodine receptor is localized on not only excitable cells such as muscle and nerve cells, but also non-excitable cells such as BMMs [86]. Ryanodine has an inhibitory effect on osteoclast function [74].

Inositol 1,4,5-trisphosphate (IP_3_) produced by phospholipase C (PLC) binds to IP_3_ receptors and releases Ca^2+^ from the ER to the cytoplasm. Although Ca oscillations do not occur in IP_3_ receptor type 2 KO mice, osteoclasts are generated. There is a complementary differentiation pathway that is independent of the Ca^2+^-NFATc1 axis [75].

Sarco/endoplasmic reticulum Ca^2+^ ATPase (SERCA) is a calcium uptake pump on the ER. In SERCA2 heterozygote mice, RANKL-induced Ca oscillations do not occur and osteoclastogenesis is diminished [76].

Transmembrane (Tmem) proteins on the ER membrane are novel therapeutic targets for bone loss. For example, Tmem178, which is down-regulated by PLCγ2, is a negative regulator of Ca oscillations and osteoclastogenesis through modulation of the NFATc1 axis [77]. Meanwhile, Tmem64 is a positive regulator of osteoclastogenesis via SERCA2-dependent Ca^2+^ signaling [78].

Ca^2+^ release-activated Ca^2+^ (CRAC) channels are composed of plasmalemmal Orai1 and Stim1 on the ER membrane. Direct influx from extracellular domain into ER through CRAC channels occurs after RANKL stimulation [32,79,87]. CRAC channels are necessary in the late phase for cell fusion to produce multinucleated osteoclasts [80]. Another group showed CRAC channels are necessary for NFATc1 activation in the early phase of osteoclast differentiation [81]. Orai1-depleted mice exhibit skeletal impairments through inhibition of osteoclast and osteoblast differentiation [82,88]. Stim1 mutations are associated with immunodeficiency and dentition defects [89].

### 4.2. Lysosome and Mitochondria, and Nucleus

Lysosomes and mitochondria are typical Ca^2+^ storage organelles. Lysosomal Ca^2+^ release mediated by TRP family member TRPML1 is necessary for osteoclastogenesis [83]. Mitochondrial granules in bone-resorbing osteoclasts contain abundant Ca^2+^ [90].

The nucleus is another Ca^2+^ storage organelle. In stimulated osteoclasts, nuclear Ca^2+^ increases in a similar manner to cytosolic Ca^2+^ [84], and integrin receptors mediate the intranuclear Ca^2+^ concentration [85].

## 5. Environmental Factors Affecting Intracellular Calcium of Osteoclast

Ca channels, transporters, and intracellular organs are major components inducing Ca oscillations. However, we cannot miss environmental factors when explaining Ca oscillations. In this section, we will discuss on important outsiders affecting intracellular Ca concentration.

Measurement methods of Ca oscillations in Section 5 are summarized in Table 4.

### 5.1. Extracellular Calcium and Calcium-Sensing Receptor

The extracellular free Ca^2+^ concentration is much higher than the intracellular concentration, especially in the bone microenvironment. Although there is a large difference between the Ca^2+^ concentrations inside and outside cells, intracellular Ca^2+^ is maintained within a narrow range, and subtle intracellular Ca^2+^ changes are commonly used for second messenger signaling. In osteoclasts, intracellular Ca oscillations are known to affect their differentiation and function according to the surrounding environment.

The extracellular Ca^2+^ concentration also affects osteoclast differentiation and function. Zaidi et al. reported that the high Ca^2+^ concentration at bone resorption sites led to a high Ca^2+^ concentration in osteoclasts and directly limited osteoclast function [91]. In contrast, other reports described that a high Ca^2+^ concentration stimulated osteoclast differentiation and bone-resorption function in a coculture system with osteoblasts [96,97,98]. The extracellular Ca^2+^ concentration also affects osteoclast migration [99]. Interestingly, Xiang et al. showed osteoclasts became attached to the bone surface when the Ca^2+^ concentration was low [100]. Furthermore, the external Ca^2+^ concentration has effects on osteoclast survival [101] and apoptosis [102].

Calcium-sensing receptor (CaSR) is a major transducer of information on the extracellular Ca^2+^ concentration to osteoclast precursors and mature osteoclasts. A high Ca^2+^ concentration directly promotes osteoclastogenesis via CaSR [103]. CaSR is also present in mature osteoclasts [104]. In bone growth plate maturation, CaSR is necessary for 25-hydroxyvitamin D-1α-hydroxylase, which modulates systemic Ca^2+^ concentration, to evoke calcium-stimulated bone erosion [105]. In addition, CaSR is involved in osteoclast apoptosis [106,107]. The downstream of CaSR contains the phosphoinositide 3-kinase/Akt pathway [108] and RANKL signaling pathway [92].

The systemic serum Ca^2+^ level is maintained by hormonal regulation, including parathyroid hormone and vitamin D. Calcitonin secreted by the thyroid gland also regulates systemic Ca^2+^. Calcitonin inhibits bone resorption and Ca^2+^ release from bone through calcitonin receptors on osteoclasts [109]. In addition, calcitonin-induced Ca^2+^ decreases affect osteoclast shape and bone resorption [110,111]. In a recent paper, calcitonin was shown to induce bone formation by interrupting sphingosine 1-phosphate release from osteoclasts [112].

### 5.2. Protons and Reactive Oxygen Species

The extracellular proton level measured by pH is the simplest factor involved in the intracellular Ca^2+^ concentration. Acid-sensing ion channel 1a promotes acid-induced osteoclastogenesis [93]. Kato and Matsushita showed that protons contributed to osteoclast and osteoblast differentiation independently of bicarbonate ions [113].

In the acidified situation, osteoclasts decrease their cytosolic Ca^2+^, synthesize extracellular matrix [94], and promote bone resorption [114]. In addition, accumulation of acids and Ca^2+^ acts as negative feedback for vacuolar-type H^+^-ATPase [115].

Another environmental factor involved in the fine-tuning of Ca oscillations is oxidative stress. Reactive oxygen species promotes osteoclast differentiation via NF-κB activation and Ca^2+^ efflux from endoplasmic reticulum [116]. RANKL induced reactive oxygen species production and enduring Ca oscillations [87]. Osteoclast differentiation decreased with the disruption of oxidative stress and Ca^2+^ signaling by asperpyrone A [95].

## 6. Perspectives for Research on Ca Oscillations in Osteoclast

In this review, we have described the key players that influence intracellular Ca oscillations. In the last section, we will focus on unsolved questions around Ca oscillations in osteoclast differentiation, function, and apoptosis.

### 6.1. No Consensus for the Definition of Ca Oscillations or Spikes

There is no common definition of Ca oscillations. This is the most critical problem associated with discussing Ca oscillations in osteoclasts. Many papers have shown line graphs for fluorescence ratios of Ca^2+^ indicators, Fura-2, or Fluo-4 and Fura Red. Qualitative assessment of Ca oscillations by appearance has been separately performed by the authors and their readers. Although some papers tried quantitative assessment with frequency and amplitude of spike or peak, these methods are in a minority (tables).

A lack of intracellular Ca^2+^ changes is not a physiological condition for viable cells. However, Ca oscillations were often binarized and regarded as all-or-none phenomena. Knockout of specific molecules often abolished Ca oscillations. Ca oscillations tended to be underestimated, and subtle changes in oscillations were omitted.

Our group proposed a new analytical method for Ca oscillations in a recent paper [21]. Concretely speaking, time-series Ca oscillations can be transformed into a frequency domain and evaluated in terms of speed. Fast Fourier transformation and wavelet analysis may be useful for detecting quick responses to environmental stimuli. A gold standard for Ca oscillation analysis should be established with modern computer technology and physio-mathematical methods.

### 6.2. Ca oscillation Alterations According to the Differentiation Time Course

Another problem is the variation in focused timing of osteoclastogenesis in different research. Experimental setting differs by each research (tables). In addition, Ca oscillations have not been well examined in the early phase of osteoclast differentiation, because Ca^2+^ signaling is considered weak in the immature phase. However, even in BMMs or osteoclast precursor cells, spontaneous Ca oscillations can be observed [21]. The Ca oscillation changes during the time course of osteoclast differentiation from BMMs to mature osteoclasts should be discussed in detail.

### 6.3. Whether Macrophage Colony-Stimulating Factors (M-CSF) or RANKL Can Evoke Ca Oscillations?

Surprisingly, it remains unclear whether RANKL can evoke Ca oscillations in electrically stable cells such as BMMs. Though some papers showed Ca^2+^ spike by acute RANKL stimulation, some research did not.

In our experiments, RANKL did not induce rapid responses to intracellular Ca oscillations [21]. It is difficult to separate the effect of RANKL on Ca oscillations from those of other environmental factors including extracellular Ca^2+^ itself, pH, temperature, and macrophage colony-stimulating factors (M-CSF; CSF-1), as other stimuli required for osteoclastogenesis. Accumulation of well-controlled observations would solve the simple but profound question of the contribution ratio of each component.

### 6.4. Direct Relationships between ITAM Receptors and Ca^2+^ Channels, Transporters, and Storage Organelles?

Immunoreceptor tyrosine-based activation motifs (ITAMs) are regarded as necessary costimulatory signals during osteoclastogenesis [15]. However, it remains unclear how costimulatory signals including downstream phosphorylation of Syk and PLC are directly linked to quick Ca oscillations and calcineurin activity changes. The frequency conversion mechanism for costimulatory signals should be examined with modern technology. For example, a visualization technique for the phosphorylation status of single molecules would provide clues. Appropriate usage of calcium indicators and imaging techniques is considerably important.

### 6.5. Identification of the Conductor of Finely-Tuned Ca Oscillations and Clarification of the True Causal Relationship between Ca Oscillations and Osteoclast Differentiation

Previous papers have shown that many molecules are involved in fine-tuning of Ca oscillations in a cooperative manner. Ca oscillations in osteoclastogenesis appear to act in a coordinated and orchestrated manner through proper regulation of calcium channels, pumps, and intracellular organelles. However, we do not have evidence on the mechanism for the integrated coordination of the components and their contributions to the harmony of the process. This is partly because Ca oscillations in osteoclastogenesis have mainly been examined in deteriorating, rather than physiological, situations in some KO mice. To reveal the underlying mechanism for the orchestration, novel approaches other than the use of single inhibitors or knockdown may be necessary.

Furthermore, we cannot reach a conclusion on whether Ca oscillations are a cause or a result of osteoclastogenesis. If we can create proper Ca^2+^ changes and accomplish the induction of osteoclasts without M-CSF and RANKL stimuli, Ca oscillations may be proven as a cause of osteoclast differentiation. However, we cannot realize such artificial osteoclast induction with the currently available technology.

## 7. Conclusions

In osteoclast differentiation, many actors play to maintain Ca oscillations (Figure 1). Recent studies showed that the balance and the strength of costimulatory signals producing optimal Ca oscillations are important. Ca^2+^ channels and transporters work in harmony to alter intracellular Ca^2+^ concentration. Transient receptor potential (TRP) channels on cell membrane are key tuners of Ca oscillations in adaptation to the environment.

Osteoclast differentiation has not yet been resolved, because the fundamental mechanism underlying the well-known phenomena of Ca oscillations remains unclear. There may be hidden therapeutic target points in the electrophysiological regulation of osteoclast differentiation and function. Further investigations on Ca oscillations will lead to comprehensive understanding of osteoclasts.

## Figures and Tables

**Figure 1 ijms-22-00180-f001:**
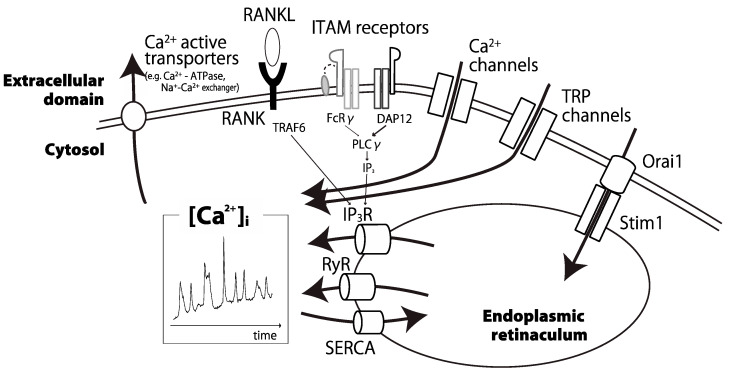
Schematic representation of the main modifiers of calcium oscillations in osteoclasts. Some calcium channels, pumps, ITAM receptors, and intracellular organelles coordinate intracellular calcium oscillations in a proper manner. [Ca^2+^]_i_: intracellular calcium ion concentration; RANK: receptor activator of nuclear factor kappa B; RANKL: RANK ligand; ITAM: immunoreceptor tyrosine-based activation motif; FcRγ: Fc receptor gamma; DAP12: DNAX-activating protein of 12 kD; TRP: transient receptor potential; IP_3_: inositol 1,4,5-trisphosphate; RyR: ryanodine receptor; SERCA: sarco/endoplasmic reticulum Ca^2+^ ATPase.

**Table 1 ijms-22-00180-t001:** Ca oscillations studies related to costimulatory signals in osteoclast differentiation.

Focused Molecules or Organs	Main Effect on Ca Oscillations	Animal or Cell Line	Pretreatment Condition or Cell Type	Reagents for Ca^2+^ Measurement	Measurement Interval (s)	Assessment of Ca Oscillations	First Author	Year	
NFATc1, RANKL, IL-1	IL-1, suppressive	Mouse	24–72 h RAKNL & M-CSF	ratiometry, Fluo-4 & Fura Red	5	by appearance	Takayanagi	2002	[8]
ITAM, costimulatory signals	Dap12 KO, suppressive	Mouse	24 h RANKL & M-CSF	ratiometry, Fluo-4 & Fura Red	5	by appearance	Koga	2004	[15]
protein tyrosine kinase (PTK)	PTK inhibitors, [Ca^2+^]_i_ level↑	Rat	Osteoclast	ratiometry, Fura-2	2 to 3	by appearance	Kajiya	2000	[18]
IL-10	IL-10, suppressive	Human	24 h RANKL	ratiometry, Fura-2	15	by appearance	Park-Min	2009	[19]
immune complex, FcRγ	Fcgr3 KO, promotive	Mouse	BMM	ratiometry, Fluo-4 & Fura Red	5	by appearance	Negishi-Koga	2015	[20]
CTLA4, FcRγ	CTLA4-Ig, suppressive	Mouse	BMM	ratiometry, Fura-2	2 to 3	difference, wavelet method	Okada	2019	[21]

ITAM: immunoreceptor tyrosine-based activation motif; FcRγ: Fc receptor gamma; CTLA4: cytotoxic T-lymphocyte antigen 4; KO: knock out; BMM: bone marrow macrophage.

**Table 2 ijms-22-00180-t002:** Ca oscillations studies related to Ca^2+^ channels and transporters.

Focused Molecules or Organs	Main Effect on Ca Oscillations	Animal or Cell Line	Pretreatment Condition or Cell Type	Reagents for Ca^2+^ Measurement	Measurement Interval (s)	Assessment of Ca Oscillations	First Author	Year	
TRPV2, Stim1, Orai1	Trpv2 KD, suppressive; Stim1 KD, suppressive; Orai1 KD, suppressive	RAW cell	18, 48 h RANKL	ratiometry, Fura-2	2 to 3	oscillation frequency	Kajiya	2010	[32]
TRPV2, multiple myeloma (MM)	Trpv2 overexpression, Ca^2+^ influx faster	RAW, MM cell	response to outcellular Ca	normalized intensity, Fluo-4	5	response curve	Bai	2018	[33]
TRPV4, Stim1	Trpv4 KD, oscillation peak↓; Stim1 KD, oscillation peak↓	RAW cell	4, 8 day RANKL & M-CSF	normalized intensity, Fluo-4	1.5	peak number, time to peak	Li	2018	[34]
TRPV4	TRPV4 activation, Ca^2+^ influx↑	Mouse	5 day RANKL & M-CSF	ratiometry, Fura-2	2 to 3	%oscillations, peak frequency, amplitude	Masuyama	2008	[35]
TRPV4	TRPV4 activation, Ca^2+^ influx↑	Mouse	Osteoclast	ratiometry, Fura-2	2 to 3	by appearance	Masuyama	2012	[36]
TRPV5	TRPV5 KD, no RANKL-induced [Ca^2+^]_i_ elevation	Human	acute RANKL stimulation to Osteoclast	ratiometry, Fura-2	2	intracellular Ca concentration change	Chamoux	2010	[37]
TRPV6	Trpv6 KO, no change	Mouse	72 h M-CSF	ratiometry, Fluo-4 & Fura Red	5	by appearance	Chen	2014	[38]
TRPC6, TRPC3	Trpc6 KD, [Ca^2+^]_i_ level↑; TRPC3 inhibition [Ca^2+^]_i_ level↓	RAW cell	1 day RANKL	ratiometry, Fura-2	<2	intracellular Ca concentration change	Klein	2020	[39]
cation sensitive receptors	Ni^2+^, [Ca^2+^]_i_ level↑; K^+^ ionophore, [Ca^2+^]_i_ level↓	Rat	Osteoclast	ratiometry, Fura-2	1	by appearance	Pazianas	1993	[40]
T-type Ca^2+^ channel Cav3.2	Cav3.2 inhibition, suppressive	Mouse	3 day RANKL & M-CSF	ratiometry, Fluo-4 & Fura Red	10	by appearance	Koide	2009	[41]
voltage-gated Ca^2+^ channel	voltage-gated Ca^2+^ channel activation, [Ca^2+^]_i_ level↑	Chicken	Osteoclast	ratiometry, Fura-2	<2	by appearance	Miyauchi	1990	[42]
RGS12	Rgs12 KD, suppressive	Mouse, RAW cell	24, 48, 72 h RANKL & M-CSF	ratiometry, Fluo-4 & Fura Red	5	by appearance	Yang	2007	[43]
RGS12	Rgs12 KO, suppressive	Mouse	24 h RANKL & M-CSF	intensity, Fluo-4	5	by appearance	Yuan	2015	[44]
RGS10	Rgs10 KO, suppressive	Mouse	72 h RANKL & M-CSF	ratiometry, Fluo-4 & Fura Red	5	by appearance	Yang	2007	[45]
Ca^2+^-activated K^+^ channel KCa3.1	KCa3.1 inhibition, RANKL-induced [Ca^2+^]_i_ change↓	Mouse	acute RANKL stimulation to BMM	normalized intensity, Fluo-4	1	%response cells, amplitude	Grossinger	2018	[46]
K^+^ channel subfamily K member 1 (KCNK1)	KCNK1 overexpression, suppressive; high [K^+^]_o_, suppressive	Mouse	48 h RANKL & M-CSF	ratiometry, Fura-2	2 to 3	by appearance	Yeon	2015	[47]
membrane potential change via K^+^ channels	high [K^+^]_o_ & [Ca^2+^]_i_ high -> [Ca^2+^]_i_ level↓, high [K^+^]_o_ & [Ca^2+^]_i_ low -> [Ca^2+^]_i_ level↑	Rat	Osteoclast	ratiometry, Fura-2	2 to 3	by appearance	Kajiya	2003	[48]
plasma membrane Ca^2+^-ATPase (PMCA)	PMCA KD, promotive	Mouse	2 day RANKL & M-CSF	ratiometry, Fura-2	0.5	by appearance	Kim	2012	[49]
Na^+^-Ca^2+^ exchanger (NCX)	NCX inhibitors, [Na^+^]_o_-free-induced [Ca^2+^]_i_ increase↓	Mouse	Osteoclast	ratiometry, Fura-2	<5	relative change of ratio, rate of change	Li	2007	[50]

TRPV: TRP vanilloid; TRPC: TRP canonical; RGS: regulator of G protein signaling; KO: knock out; KD: knock down; BMM: bone marrow macrophage.

**Table 3 ijms-22-00180-t003:** Ca oscillations studies related to intracellular calcium storage.

Focused Molecules or Organs	Main Effect on Ca Oscillations	Animal or Cell Line	Pretreatment Condition or Cell Type	Reagents for Ca^2+^ Measurement	Measurement Interval (s)	Assessment of Ca Oscillations	First Author	Year	
Ryanodine, Ruthenium Red	Ryanodine, [Ca^2+^]_i_ level↑; Ruthenium Red, [Ca^2+^]_i_ level↓	Rat	Osteoclast	ratiometry, Fura-2	~10	by appearance	Ritchie	1995	[74]
inositol 1,4,5-trisphosphate receptor (IP_3_R)	IP_3_R type 2,3 KO, suppressive	Mouse	48–72 h RANKL & M-CSF	ratiometry, Fura-2	<5	by appearance	Kuroda	2008	[75]
Sarco/endoplasmic reticulum Ca^2+^ ATPase (SERCA)	SERCA2 +/−, BMM: initial peak↓& preOC: spikes↓	Mouse	BMM, 48 h RANKL	ratiometry, Fura-2	2	peak ratio, spike frequency	Yang	2009	[76]
Transmembrane(Tmem)178	Tmem 178 KO, [Ca^2+^]_i_ level↑	Mouse	BMM, pre osteoclast	ratiometry, Fura-2	2	by appearance	Decker	2015	[77]
Tmem64, SERCA	Tmem64 KO, suppressive	Mouse	48 h RANKL & M-CSF	ratiometry, Fluo-4 & Fura Red	5	by appearance	Kim	2013	[78]
Stim1	Stim1 mutations, Store-operated Ca^2+^ entry↓& M-CSF+RANKL-induced [Ca^2+^]_i_ elevation↓	Human	BMM	ratiometry, Fura-2	<2	by appearance	Huang	2020	[79]
CRAC channel	RANKL stimulation, Ca^2+^ influx longer	Human	0, 1, 3, 7, 11 day RANKL & M-CSF	ratiometry, Fura-2	1.5	%oscillation cells, average Ca entry	Zhou	2011	[80]
Orai1	Orai1 KD, [Ca^2+^]_o_↑→[Ca^2+^]_i_ elevation↓	RAW cell	RAW cell	ratiometry, Fura-2	<10	peak value, initial rate of rise	Hwang	2012	[81]
Orai1	Orai1 KO, [Ca^2+^]_o_↑→[Ca^2+^]_i_ elevation↓	Mouse	BM-derived stromal cells	ratiometry, Fura-2	<10	by appearance	Hwang	2012	[82]
TRPML1, Lysosome	TRPML1 KO, spike number & amplitude↓	Mouse	48 h RANKL	ratiometry, Fura-2	<10	spike frequency, by appearance	Erkhembaatar	2017	[83]
Nuclear, Cytosolic	ATP, [Ca^2+^]_i_ level↑, Integrin-binding peptide, [Ca^2+^]_i_ level↑,	Rat	Osteoclast	ratiometry, Fura-2	<2	by appearance	Parkinson	1998	[84]
Nuclear, integrin receptor	integrin ligands, [Ca^2+^]_i_ level↑	Rat	Osteoclast	ratiometry, Fura-2	<3	by appearance	Shankar	1993	[85]

KO: knock out; KD: knock down; BMM: bone marrow macrophage; BM: bone marrow.

**Table 4 ijms-22-00180-t004:** Ca oscillations studies related to environmental factors.

Focused Molecules or Organs	Main Effect on Ca Oscillations	Animal or Cell Line	Pretreatment Condition or Cell Type	Reagents for Ca^2+^ Measurement	Measurement Interval (s)	Assessment of Ca Oscillations	First Author	Year	
extracellular Ca^2+^	[Ca^2+^]_o_↑, [Ca^2+^]_i_ level↑	Rat	Osteoclast	ratiometry, indo-1	<5	by appearance	Zaidi	1989	[91]
extracellular Ca^2+^	[Ca^2+^]_o_↑, [Ca^2+^]_i_ level↑	RAW cell	Osteoclast	ratiometry, Fura-2	<10	by appearance	Xu	2005	[92]
Acid-sensing ion channel (ASIC) 1a	acid, [Ca^2+^]_i_ level↑, acid & ASIC1a inhibition, [Ca^2+^]_i_ level↑diminished	Rat	Osteoclast	ratiometry, Fura-2	<5	amplitude, by appearance	Li	2013	[93]
extracellular proton (pH)	acid, [Ca^2+^]_i_ level↑	Rat	Osteoclast	ratiometry, Fura-2	<5	by appearance	Teti	1989	[94]
oxidative stress, asperpyrone A (antioxidant)	asperpyrone A, suppressive	Mouse	24 h RANKL	intensity, Fluo-4	2	intensity change	Chen	2019	[95]
Reactive Oxygen Species (ROS)	peroxiredoxin(Prx) II KO (ROS↑), promotive	Mouse	BMM, 48 h RANKL	ratiometry, Fura-2	<10	spike frequency	Kim	2010	[87]

KO: knock out; KD: knock down; BMM: bone marrow macrophage; BM: bone marrow.

## Data Availability

Data sharing not applicable.

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
