# Peer review of "Finely-Tuned Calcium Oscillations in Osteoclast Differentiation and Bone Resorption"

_ijms, 2020, doi:10.3390/ijms22010180_

Round 1
Reviewer 1 Report
Okada and colleagues made a great deal of work to synthesize a quite important amount of literature in order to give a clear review of the knowledge concernning calcium oscillations in osteoclast differentiation and function. This should be acknoledge, and this is very interesting to the bone field.
Although this review will take place in a special issue about updates of calcineurin/NFAT signaling in human health and disease, it would probably be of interest to quickly remind the reader of the importance of Ca in the NFATc1 – calcineurine just in a little more details than what is already done.
There are few English mistakes that has to be corrected.
The review is really interresting and should be accept for publication.
Author Response
We thank the reviewers for thoughtful evaluation of our manuscript. We have given careful attention to their suggestions and addressed their comments. Point-by-point responses to specific comments are provided below. The reviewers’ comments are in bold and italic. The changes made in the manuscript were written in red.
To Reviewer 1
Okada and colleagues made a great deal of work to synthesize a quite important amount of literature in order to give a clear review of the knowledge concerning calcium oscillations in osteoclast differentiation and function. This should be acknowledged, and this is very interesting to the bone field.
Although this review will take place in a special issue about updates of calcineurin/NFAT signaling in human health and disease, it would probably be of interest to quickly remind the reader of the importance of Ca in the NFATc1 – calcineurin just in a little more details than what is already done.
→ Thank you for your valuable comments. According to your advice, we added further description of calcineurin & Ca oscillations in osteoclast differentiation.
We described how calcineurin is important in osteoclast differentiation mediating Ca oscillations signal to NFATc1 activation. In abstract, we added the sentence “Calcium oscillations (Ca oscillations) are well-known phenomena in receptor activator of nuclear factor kappa B ligand (RANKL)-induced osteoclastogenesis and bone resorption via calcineurin.”
There are few English mistakes that has to be corrected.
→ We apologize for bothering you in reading manuscripts. We reviewed expressions and phrases in the manuscript, and corrected them if necessary.

Reviewer 2 Report
In the present manuscript, the authors have compiled results from studies on calcium oscillations during osteoclast differentiation. There are many different mechanisms by which intracellular calcium is regulated through both extra- and intracellular compartments. For this reason, the manuscript is divided in seven parts, called “chapters” by the authors. I suggest “sections” would be a more appropriate term. It is an impressive amount of observations presented in the manuscript. The manuscript has a tendency, however, to be a enumeration of findings although they are nicely presented in different sections. This makes it different for a reader to get an overall view of the topic. At the end, the authors have a “Conclusion” section but this does not help very much. I suggest the authors try to summarize the observations, or the most important observations, in the last section, preferable together with a schematic figure.
Specific comments:
- In the abstract, line 20 is stated that FcRgamma has a role in osteoclast differentiation only in inflammatory situations. This statement is difficult to reconcile with the fact that both DAP12 and FcRgamma are required for RANKL induced osteoclastogenesis in bone marrow macrophage cultures and that FcgammaRIII inhibits osteoclastogenesis during physiological conditions according to Negishi-Koga et al.
- Page 2, line 54: osteoclast should be osteoclasts
- Page 2, section/chapter 2: the redundancy of DAP12 and FcRgamma should be mentioned
- Page 3, lines 111-113: I suggest the authors describe the findings by Negishi-Kota et al in more details.
- Page 4, line 125: signal should be signals
- Page 4, line 126: does should be do
- Page 5, line 159: I suggest “and reduces bone resorption” is described in more detail, for example by stating “which acts in a negative feed back mechanism to reduce the bone resorptive activity of mature osteoclasts”
- Page 6, lines 232-233: The observations in ref. 79 and 80 can not be reconciled. In ref. 79 is shown that downregulation of Orai1 results in decreased osteoclast formation with no effect on TRAP expression, leading the authors to conclude that CRAC channels are involved in osteoclast progenitor cell differentiation at late stages/fusion to multinucleated cells. In ref. 80 is also shown that downregulation of Orai1 decreases osteoclast differentiation but that this is associated with downregulation of NFATc1 as well as downstream ostoclastic genes such as TRAP and cathepsin K, which shows that CRAC channels are important also for early steps during osteoclast differentiation and not only at terminal steps.
- Page 7, line 273: it should be described how CaSR interacts with 25-(OH)-1alpha-hydroxyase at the growth plate
- Page 9, conclusion: it remains many more unresolved questions than only the fundamental mechanisms underlying Ca osillations before we have a full picture of osteoclast differentiation in physiological and pathological processes.
Author Response
We thank the reviewers for thoughtful evaluation of our manuscript. We have given careful attention to their suggestions and addressed their comments. Point-by-point responses to specific comments are provided below. The reviewers’ comments are in bold and italic. The changes made in the manuscript were written in red.
In the present manuscript, the authors have compiled results from studies on calcium oscillations during osteoclast differentiation. There are many different mechanisms by which intracellular calcium is regulated through both extra- and intracellular compartments. For this reason, the manuscript is divided in seven parts, called “chapters” by the authors. I suggest “sections” would be a more appropriate term.
→ Thank you for your comments. I changed the nomenclature “chapters” to “sections”.
It is an impressive amount of observations presented in the manuscript. The manuscript has a tendency, however, to be a enumeration of findings although they are nicely presented in different sections. This makes it difficult for a reader to get an overall view of the topic.
At the end, the authors have a “Conclusion” section but this does not help very much. I suggest the authors try to summarize the observations, or the most important observations, in the last section, preferable together with a schematic figure.
→ Thank you for your advice. In the last section, we added one paragraph to summarize the knowledge of main tuners of Ca oscillations with a quotation of figure 1. We emphasized two topics in Ca oscillations of osteoclast. The balance and the strength of costimulatory signals are important in osteoclast differentiation from recent studies. TRP channels are key regulators for adaptation to the environment.
Specific comments:
- In the abstract, line 20 is stated that FcRgamma has a role in osteoclast differentiation only in inflammatory situations. This statement is difficult to reconcile with the fact that both DAP12 and FcRgamma are required for RANKL induced osteoclastogenesis in bone marrow macrophage cultures and that FcgammaRIII inhibits osteoclastogenesis during physiological conditions according to Negishi-Koga et al.
→ Thank you for your valuable comment. As you mentioned, the expression “FcRγ has a role in osteoclast differentiation only in inflammatory situations” is exaggerated, because in physiological situation, FcRγ couples with PIR-A or OSCAR and affects osteoclastogenesis to some extent. We corrected this sentence as below “FcRγ works as a key controller of osteoclastogenesis especially in inflammatory situation.”
- Page 2, line 54: osteoclast should be osteoclasts
→ Thank you for your suggestion. We corrected osteoclast to the plural form in line 54.
- Page 2, section/chapter 2: the redundancy of DAP12 and FcRgamma should be mentioned
→ Thank you for your comment. To slim down the description of DAP12-related molecules, we deleted these sentences, “In the downstream of Siglec-15, osteoclast differentiation is affected by the RANKL-induced phosphatidylinositol 3-kinase/Akt and Erk pathways [21]. Siglec-15 neutralizing antibodies are potential drugs for osteoporosis [22] [23], ” because down-stream of Siglec-15 is not necessary in the paragraph focusing on DAP12.
- Page 3, lines 111-113: I suggest the authors describe the findings by Negishi-Kota et al in more details.
→ Thank you for your suggestion. According to your advice, we added the description of FcRγ signaling modified by FcγR coupling patterns studied by Dr. Negishi-Koga as below “In the physiological state, FcRγ couples with PIR-A or OSCAR and promotes osteoclastogenesis. While, in the pathological inflammatory situation with an abundance with immune complex, FcRγ binds to several types of Fcγ receptors. FcγR I/III/IV strengthen FcRγ signaling, in contrast, FcγRIIB weaken FcRγ signaling. The coupling patterns of Fcγ receptors with FcRγ modify the strength of FcRγ signaling and affect osteoclastogenesis.”
- Page 4, line 125: signal should be signals
- Page 4, line 126: does should be do
→ Thank you for your kind suggestion. We corrected signal to signals, does to do.
- Page 5, line 159: I suggest “and reduces bone resorption” is described in more detail, for example by stating “which acts in a negative feedback mechanism to reduce the bone resorptive activity of mature osteoclasts”
→ Thank you for your comment. According to your suggestion, we changed the sentence as below. “TRPV5 mediates RANKL-induced intracellular Ca2+ increases and reduces bone resorption due to a negative feedback mechanism to reduce the bone resorptive activity of mature osteoclasts.”
- Page 6, lines 232-233: The observations in ref. 79 and 80 cannot be reconciled. In ref. 79 is shown that downregulation of Orai1 results in decreased osteoclast formation with no effect on TRAP expression, leading the authors to conclude that CRAC channels are involved in osteoclast progenitor cell differentiation at late stages/fusion to multinucleated cells. In ref. 80 is also shown that downregulation of Orai1 decreases osteoclast differentiation but that this is associated with downregulation of NFATc1 as well as downstream osteoclastic genes such as TRAP and cathepsin K, which shows that CRAC channels are important also for early steps during osteoclast differentiation and not only at terminal steps.
→ Thank you for your comment. We apologized the confusing description of the CARC channels’ role. To clarify the important phase for CRAC channels in osteoclastogenesis, we divided the description as below. “CRAC channels are necessary in the late phase for cell fusion to produce multinucleated osteoclasts [79]. Another group showed CRAC channels are necessary for NFATc1 activation in the early phase of osteoclast differentiation [80].
- Page 7, line 273: it should be described how CaSR interacts with 25-(OH)-1alpha-hydroxyase at the growth plate
→ Thank you for your advice. 25-(OH)-1alpha-hydroxyase (1alpha(OH)ase) works as a regulator of systemic Ca2+concentration. Richard et al. performed histomorphometrically study comparing femur from 1alpha(OH)ase knock out mice and 1alpha(OH)ase and Casr double knock out (DKO) mice. 1alpha(OH)ase -/- and Casr -/- DKO mice showed the arrest of bone linear growth in normal diet, in contrast, 1alpha(OH)ase -/- KO mice did not show the arrest. They concluded that CaSR is necessary when systemic change of Ca2+ concentration affect maturation of the maturation of cartilaginous growth plate and bone erosion. To describe the relevance between CaSR and 1alpha(OH)ase clearly, I changed the sentence as below. “In bone growth plate maturation, CaSR is necessary for 25-hydroxyvitamin D-1α-hydroxylase, which modulates systemic Ca2+ concentration, to evoke calcium-stimulated bone erosion [98]”
- Page 9, conclusion: it remains many more unresolved questions than only the fundamental mechanisms underlying Ca oscillations before we have a full picture of osteoclast differentiation in physiological and pathological processes.
→ Thank you for your comments. We agreed with your opinion. Subtype of osteoclast is the latest topic in osteoclast research, however, the topic of pathological osteoclast has a little relationship with the mechanism of physiological Ca oscillations. We deleted section 6.5.
